# Synergy Masks of Domain Attribute Model DaBERT: Emotional Tracking on Time-Varying Virtual Space Communication

**DOI:** 10.3390/s22218450

**Published:** 2022-11-03

**Authors:** Ye Wang, Zhenghan Chen, Changzeng Fu

**Affiliations:** 1College of Computer, National University of Defense Technology, Changsha 410073, China; 2School of Software & Microelectronics, Peking University, Beijing 100191, China; 3SSTC, Northeastern University, Qinhuangdao 066004, China; 4Graduate School of Engineering Science, Osaka University, Toyonaka 560-0043, Osaka, Japan

**Keywords:** emotional tracking, virtual communication, pre-trained task, domain attribute

## Abstract

Emotional tracking on time-varying virtual space communication aims to identify sentiments and opinions expressed in a piece of user-generated content. However, the existing research mainly focuses on the user’s single post, despite the fact that social network data are sequential. In this article, we propose a sentiment analysis model based on time series prediction in order to understand and master the chronological evolution of the user’s point of view. Specifically, with the help of a domain-knowledge-enhanced pre-trained encoder, the model embeds tokens for each moment in the text sequence. We then propose an attention-based temporal prediction model to extract rich timing information from historical posting records, which improves the prediction of the user’s current state and personalizes the analysis of user’s sentiment changes in social networks. The experiments show that the proposed model improves on four kinds of sentiment tasks and significantly outperforms the strong baseline.

## 1. Introduction

Nowadays, social networks have become the main communication platform for people: more and more people are expressing their opinions and feelings on social network platforms. Social networks allow users to break through the geographical distance and have become an indispensable part of people’s lives. Social networks also break the boundaries of time, and recording historical information enables people in different time zones to communicate with each other. Emotional tracking on time-varying virtual space communication is a social network mining approach that collects and analyzes opinions about different content that appears in blog posts, reviews, or tweets.

Despite the increased use of sentiment analysis for mining sentiment and options on products, services, and public interests, existing methods often focus on single posts instead of studies on temporal sentiment analysis [1]. Since social networks have time-varying properties, sentiment analysis based on time series prediction helps in opinion monitoring, financial forecasting, and business decision making. Traditional time-varying forecasting models are mainly based on statistical science, and representative methods, such as Autoregressive Integrated Moving Average (ARIMA) [2], are widely used in economics, finance, and sociology research. In recent years, some researchers have introduced deep neural network models into the study of time series prediction, bringing new research ideas to time series prediction. Related research uses sentiment analysis models to extract sentiment attributes at different time points, so as to obtain the time series of sentiment attributes, and then uses temporal prediction models to analyze the process of sentiment change. Social networks are time-sensitive, and we view user’s posts over time as a time series. In this article, we take a global perspective and integrate users’ current moment postings and historical posting records for personalized sentiment time series analysis.

In order to extract the user’s emotional style and the inner pattern of emotional change, we propose a domain-knowledge-enhanced pre-trained encoder based on BERT [3] to incorporate external domain knowledge, such as the emotional properties of words. BERT has advanced the state-of-the-art on the sentiment analysis task. The sentiment property of text in sentiment analysis is of great importance, and the original BERT only uses the semantic information of text itself in pre-training. Then, we predict the trend of user’s sentiment change combined with the temporal prediction model. In addition, we propose a sentiment style module based on a self-attention mechanism to extract sentiment style attributes from historical text data.

We construct a new time series dataset based on Sentiment140 and demonstrate that our model has a superior ability to predict trends in sentiment change. Experiments on four datasets in English and Chinese also show that our proposed model significantly improves strong baselines for sentiment analysis.

The contributions of our work are summarized as follows:We propose a novel sentiment analysis model for an emotional state personalized analysis that incorporates a time-varying prediction approach to emotional tracking on time-varying virtual space communication. The model uses a multidimensional time-varying prediction model to mine information from the user’s historical posting records, thus extracting the user’s sentiment style and the inner pattern of sentiment change.We propose a domain-knowledge-enhanced pre-trained encoder that incorporates external knowledge of word sentiment properties into pre-trained language models. It is helpful to use the encoder to extract the representation vector of the text at each time point.Our proposed model can improve the ability to predict the user’s emotional state at the current moment. It also can also enhance the capability of semantic modeling and fine-grained differentiation, which outperform various baseline models on multi benchmark datasets.

## 2. Related Work

### 2.1. Sentiment Analysis

To obtain a representational vector of text, early sentiment analysis relied on manual feature construction based on statistical rule metrics or the researcher’s prior knowledge, and then used traditional machine learning for text classification. Pang et al. [4] disassemble the original text, create a variety of text features based on word frequency statistics, and use machine learning models such as maximum entropy classification and a support vector machine (SVM) to classify users’ comments on online platforms as positive or negative. Poria et al. [5], on the other hand, employ the LDA topic model for sentiment analysis. LDA models the original text using the bag-of-words model, obtaining a probability distribution of text-implicit topics and characterizing the text with a vector of text in the implicit topic space, which produces better results for long texts. Lu et al. [6,7,8] improve the LDA model to enhance the topic model’s textual representation. Muhammad et al. [9] develop sentiment shift metrics based on contextual information such as negation, great celebration, and decay, as well as text features such as unigrams and bigrams. Traditional sentiment analysis relies on manually designed features, and some models are only applicable to specific domains or impose strict restrictions on use due to different relevant research experiences and corpus gaps, as well as introducing a significant amount of feature engineering.

Sentiment analysis models based on neural networks have been widely used and have gradually become the mainstream of research alongside the development of neural networks, particularly deep neural networks. Akhtar et al. [10] create an integrated model with a multi-network structure to process news headlines and short social texts, resulting in finer-grained text sentiment attributes and accurate sentiment multi-classification results. McCann et al. [11] use a migration learning approach to improve the performance of the target task (sentiment analysis) using the training results of the source task (machine translation), leveraging machine translation’s rich resource of labeled data. Yao et al. [12] use a graph structure to model the corpus dataset, with words and text acting as nodes that connect edges via co-occurrence relations between points, and use a graph convolutional network (GCN) to classify nodes in the graph to extract sentiment categories from the text.

The BERT model [3], as a representative and trained language model, has recently brought about landmark changes in the field of natural language processing research, and breakthroughs have been made in several tasks, such as sentiment analysis, named entity recognition, reading comprehension, and text entailment. We incorporated external domain knowledge, such as word sentiment attributes, into the BERT-based pre-training language model.

### 2.2. Time-Varying Prediction

Traditional time series prediction relies heavily on statistical techniques, such as the simple average, moving average, exponential smoothing, ARIMA, and so on. These methods use linear models to depict time-varying states, which are highly interpretable, but they have limitations when dealing with complex system predictions and nonlinear changes.

With the advancement of computer technology, researchers have incorporated machine learning research, particularly neural networks, into temporal prediction models, and have seen significant improvements. Lai et al. [13] propose the long-and short-term time-varying network (LSTNet) model for trend prediction, which combines a CNN network and an RNN network to extract short-term local patterns and long-term change patterns from time series. Li et al. [14] propose a DCRNN model for traffic prediction that uses a traffic grid bidirectional random walk and an encoder–decoder model to capture spatially and temporally dependent information, respectively. Franceschi et al. [15] create an unsupervised machine learning task to generate a generic representation vector for time series data and improve the representation quality, portability, and practicability. For univariate temporal prediction, Oreshkin et al. [16] create a deeply stacked fully connected neural network based on a forward–backward residual connection mechanism. Sen et al. [17] propose the DeepGLO deep neural network model for high-dimensional time series data prediction, which connects the global matrix eigendecomposition model with a convolutional network and capture the local properties of each dimensional time series through additional network structures.

People’s emotional states are constantly changing, with certain periodicity and seasonality, so they can be viewed as time sequences [18]. With the rapid development of the Internet, particularly social networks, people have become more willing to share their views and opinions in the network, and these data indicate the author’s emotional state. Data on social networks are also time-varying in the sense that users’ views and comments are continuous in time, such as users’ continuous postings over time and Twitter comments at various times in popular events, so these social network data can also be considered time-varying.

Time-varying prediction has been used by researchers in sentiment analysis, as well as tasks such as business decision making, opinion analysis, and financial forecasting. Giachanou and Crestani [19] employ a sliding average method to track people’s sentiment on public topics in social networks, building a time series model from different sentiment proportions of all tweets on a public topic. O’Connor et al. [20] examine the relationship between the sentiment attributes of users’ Twitter postings and users’ perceptions of public time, employing the MPOA system to score sentiment and the sliding mean method to construct a time series of sentiment attributes, which was found to be highly correlated with changes in public opinion polls. An et al. [21] combine traditional sentiment analysis with time-varying analysis to investigate the process of people’s sentiment change on the Twitter platform, as well as the change in people’s sentiment in response to special events. Ranco et al. [22] investigate the impact of social network users’ sentiment on stock prices, constructing time-varying data on Twitter users’ sentiment toward companies and using it for financial market forecasting.

In this article, we used continuous posting records from social platforms such as Weibo and Twitter as time series and combined time-varying prediction with sentiment analysis to forecast users’ sentiment change trends. The majority of related studies employ sentiment analysis models to extract sentiment attributes at various time points and then examine the process of sentiment change using time-varying prediction. However, we take a global view of sentiment change and do not directly extract sentiment categories at different time points in the historical state, but instead mine the information of users’ historical posting records to extract users’ sentiment styles and the inner rules of sentiment change.

## 3. Methodology

We divided sentiment temporal analysis into two subtasks in this article: single text-oriented sentiment analysis and sentiment sequence-oriented temporal prediction. In this section, we define the pre-trained language model with domain knowledge enhancement and describe the time-varying prediction model to predict the change based on features extracted from the pre-trained encoder. Figure 1 illustrates the overview of our model.

### 3.1. Domain-Knowledge-Enhanced Pre-Trained Model

Pre-trained language models, as represented by BERT, have recently brought significant changes and breakthroughs to the field of NLP research. BERT uses bidirectional transformers to pretrain a large corpus, and finetunes the pretrained model on other tasks. In this article, we built on the BERT model and improved it from pre-training tasks. Text sentiment attribute information is critical in sentiment analysis, but BERT, as a general-purpose language model, uses only the text’s own data in the unsupervised pre-training task and lacks external information, such as word sentiment polarity.

We propose a pre-training task based on multi-task learning, supplementing the information of the original pre-trained language model with external domain knowledge (e.g., sentiment lexicon). It can be used for fine-grained sentiment analysis and as a sentence encoder to extract the representational vector of text at each time point in the text time series. Figure 2 shows the illustration of the domain-knowledge-enhanced pre-trained encoder.

For language model pre-training, the BERT model employs two unsupervised machine learning tasks: a masked language model (MLM) and next sentence prediction (NSP). MLM is a bidirectional language model that can learn semantic representations based on context. It replaces parts of the text sequence with a [MASK] token at random, and the model then makes predictions about the original words at those positions. For paired data, NSP is a classification task that determines whether two sentences in a text corpus are in a subordinate sentence relationship. However, sentiment analysis is limited to single texts.

The MLM is used to predict the word at [MASK] using global contextual information by performing the following processing on the raw input text: (1) to obtain a string sequence, the original text string is tokenized; (2) to form the mask candidate set, 15% of tokens are chosen at random from all locations; (3) the mask candidate set is iterated through, with an 80% probability of replacing the token at the original position with a mask token and a 10% probability of replacing the token at that position with a random token from the corpus word list.

We used the domain knowledge dictionary as external domain knowledge. A domain knowledge dictionary D={w:p},w∈W,p∈P is a mapping between words *w* and domain-specific attributes or features *p*. We created the pre-training task—domain attribute mask (DAM)—by mapping words in the domain knowledge lexicon to word sentiment attribute polarity (e.g., positive, negative, and neutral).

DAM is a special classification task that uses contextual information to predict the characteristic attributes of words in a domain knowledge at the current location. DAM is a supervised or semi-supervised machine learning task that can manually annotate text sequences or summarize domain knowledge into a domain knowledge lexicon prior to semi-supervised annotation.

We used a domain knowledge dictionary to mark tokens in the token sequence that are not in the dictionary. During training, both DAM and MLM use cross entropy loss.
(1)$$\begin{array}{cc} CrossEntropy(x,class) & =-\log\frac{\exp(x[class])}{\sum_{j}\exp\left(x\left[j\right]\right)}\\ \; & =-x\left[class\right]+\log(\sum_{j}\exp\left(x\left[j\right]\right)) \end{array}$$

We employed a joint data annotation mechanism, which means that the training data for both MLM and DAM training tasks are generated concurrently. We designed the following training methods based on the various labeling methods:**Only_MLM** performs only the MLM pre-training task.**Only_DAM** completes only the DAM pre-training task.**Mix_Separate** simultaneously performs the annotation work of DAM and MLM, and the two annotation processes are independent of each other, i.e., the words marked [MASK] may not appear in the domain knowledge lexicon.**Mix_MLM** is annotated with MLM as the primary annotation process, and MLM annotation comes first. If the masked words appear in the domain knowledge lexicon, the corresponding feature attributes are added to the DAM tag sequence.**Mix_DAM**, in contrast to MLM, replaces all words that appear in the domain knowledge lexicon with a [MASK] token. When the number of [MASK] exceeds a certain threshold, some [MASK] are chosen at random to revert to the original words.

The model parameters of the BERT part are shared by MLM and DAM, but the parameters of the prediction modules are independent, and the prediction modules of different pre-training tasks obtain the prediction sequences $P_{MLM}$ and $P_{DAM}$ of that task separately. The loss function of MLM and DAM is
(2)$$L_{MLM}=CrossEntropy\left(P_{MLM},Label_{MLM}\right)$$
(3)$$L_{DAM}=CrossEntropy\left(P_{DAM},Label_{DAM}\right)$$

The final loss function is
(4)$$L=L_{MLM}+\lambda L_{DAM}$$
where λ is the weight parameter.

The approach described above allows for the incorporation of domain knowledge into the language model, and multiple domain knowledge dictionaries can be introduced in the following ways:**Successive serial training** is performed first using the first domain knowledge lexicon for co-training and finally using the second domain knowledge lexicon on the basis of the obtained model parameters.**Simultaneous parallel training** employs multiple domain knowledge dictionaries, each corresponding to a separate DAM model, with a loss function connecting all modules.
(5)$$L=L_{MLM}+\lambda_{1}L_{DAM1}+\lambda_{2}L_{DAM2}+\cdots +\lambda_{n}L_{DAMn}$$**Multi-round mechanism** uses just one of the individual DAM modules in each training round.

### 3.2. Attention-Based Sentiment Time-Varying Prediction Model

Sentiment analysis studies are currently centered on individual texts, but because people’s emotions are complex and delicate, it can be difficult to distinguish between positive and negative aspects in a straightforward manner. Historical text data reflect the process of user sentiment change and implies an inner change pattern, which is important for sentiment analysis.

The attention mechanism for temporal sentiment recognition can be considered a special weighting mechanism: an information integration process of feature vectors in a state that combines all previous sentiment data. The following are the specifics of this article’s attention-setting mechanism for sentiment recognition. The value vector, or all feature vectors that must be integrated, represents all historical sentiment state vectors in the time series in the sentiment recognition time-series prediction task. The query vector is related to the content of the sentiment tracking recognition task. This article uses sentiment style features and social media platform text features as query vectors. Each historical sentiment state vector will have a corresponding ley vector, and the weights of the corresponding values can be determined from query and key vectors using the attention function. Key-value pairs are one-to-one correspondences. The following describes the attention calculation functions used in this study for different scenarios:(6)$$f_{dot}\left(Q_{it},K_{jt}\right)=Q_{i}^{T}K_{i}$$
(7)$$f_{general}\left(Q_{it},K_{jt}\right)=Q_{i}^{T}W_{t}K_{i}$$
(8)$$f_{perceptron}\left(Q_{it},K_{jt}\right)=v_{t}{}^{T}Leaky\_Relu\left(W_{t}Q_{i}+U_{t}K_{i}\right)$$

We set *Q* to be *i* × *t*, where *i* is the source temporal data length, and set *K* to be *j* × *t*, where *j* is the target temporal data length, and *t* is the vector dimension. It can be seen that the vectors *Q* and *K* are equal in length because both should be in the same dimensional space and need to be compared for similarity, i.e., $t_{Q}=t_{K}=t_{X}$. However, *V* is *j* × $t_{V}$, whose vector length can be different from both, where it can be considered that the key-value pairs of *K* and *V* are representations of the same data in different state spaces, and the final attention value is a matrix of *i* × $t_{V}$, which is obtained after using the $Leaky_{R}elu$ activation function. The weight coefficients, based on the coefficients on the $t_{V}$ dimensions, are weighted and summed, and, finally, the *i* attention values with $t_{V}$ dimensions are derived.

After processing social text data to create the attention weight coefficients, query and key vectors weight all of the previous temporal state data to create the integrated vector. Each attention weight in the historical text sequence measures how much that particular moment contributed to the prediction of the current sentiment state and has a precise meaning.

For a text sequence, $S=[s_{1},s_{2},\cdots ,s_{n}]$, where *n* is the length of *S*. The publication time of each text is $T=[t_{1},t_{2},\cdots ,t_{n}]$, where $t_{1}<t_{2}<\cdots <t_{n}$. Using the domain knowledge-based text encoder, we can obtain the feature vector of the text at any time.
(9)$$E=[e_{1},e_{2},\cdots ,e_{n}]$$
where the embedding dimension of $e_{i}$ is *d*.

We designed a high-dimensional temporal prediction model based on the attention mechanism. The attention mechanism, however, is not sequence-sensitive, which means that the attention-weighted results obtained by swapping the positions of two elements in a time series remain unchanged, and thus the attention mechanism cannot exploit information about the temporal properties. To address the aforementioned issue, we propose the Time-ID mechanism for obtaining the temporal attribute vector of historical texts.

In this article, two Time-ID mechanisms were designed for the current moment of text *x* and the user’s posting history sequence $S=[s_{1},s_{2},\cdots ,s_{n}]$.

**Coarse-grained order** sorts the posting records according to the time difference between them and the current moment, from closest to farthest, reflecting the posting’s sequential relationship.**Fine-grained temporal distances** are calculated separately for each text at the time of posting and now, and are marked using time-specific absolute values. This mechanism introduces the text’s absolute time interval distance.

However, the second mechanism will encounter the issue of different time distances yielding the same Time-ID, so we combined the two mechanisms to calculate the Time-ID.

In addition, to extract sentiment style attributes from historical text data, we implemented an attention-based sentiment style module. Self-attention is a sequence modeling mechanism that can obtain not only global sequence information but also local features of time series through element interaction. Because self-attention calculates the correlations between the elements in the sequence and all other elements, it can mine the two-way variation pattern in time series.

Following Shan et al. [23], the deep crossing model was used to process features such as text vectors, sentiment style attributes, and attention weighting results from historical information.

## 4. Experiments

### 4.1. Datasets

#### 4.1.1. Sentiment Analysis Based on Time Series Prediction

For sentiment time series analysis, we generated a text time series dataset for Twitter based on the Sentiment140 sentiment analysis dataset [24]. Sentiment140 contains the user names and tweeting times of tweets in the Twitter platform, from which, the Twitter text time series dataset can be built, as shown below.

To obtain each user’s posting record separately, the raw data were aggregated by user ID.The tweet records for each user were sorted by posting time.In the experiment, each newly generated record consisted of 1 tweet and *n* tweets with the most recent posting time, with *n* set to 8.

To avoid the “future data” problem in time-varying prediction, we built the data with a non-repetitive intercept, which means that each tweet is only used once and no duplicate tweets are included in any two time series records. The resulting dataset contained 43,739 records, each of which contained the current tweet, the eight most recent historical tweets at the time of posting, and the sentiment tag of the current tweet.

#### 4.1.2. Fine-Grained Sentiment Analysis

A domain knowledge lexicon was used in this article to construct word-level domain knowledge information, increasing the amount of semantic information in the model’s text representations and enabling the extraction of fine-grained sentiment attributes. As a result, we primarily used fine-grained sentiment multi-classification datasets to fine-tune the training language model. We experimented on two public datasets: SST [25] and Yelp Review-2015 [26].

SST is a publicly available dataset created by Stanford University in the United States that is widely used in sentiment analysis tasks as well as GLUE, the base task for evaluating the quality of a language model’s semantic modeling of a single text using the Language Model Rating List. The SST dataset contains 11,855 user movie reviews from Rotten Tomatoes, with two usage modes: the Root mode contains only the complete sentence data, and the All mode contains the text subsequence corresponding to the tree’s nodes in addition to the complete sentences (the text tree of each movie review obtained using Stanford constituency parser).

Yelp is the largest lifestyle review site in the United States, where users can rate and review a variety of businesses, such as restaurants, inns, car repair shops, utility providers, and others. Yelp Review-2015 is a popular public dataset for sentiment analysis in English that arose from a data challenge organized by Yelp in 2015. Yelp Review Full includes 650,000 training and 50,000 test data sets. Yelp Review Full categorizes user reviews from 1 to 5 based on negative to positive sentiment attributes. The five categories are distributed evenly, with 130,000 data for each in the training set and 10,000 data for each in the test set.

#### 4.1.3. Domain Knowledge Dictionary

The domain knowledge dictionaries used in this article include SentiWordNet3 [27] and Senticnet5 [28].

SentiWordNet is a popular English sentiment dictionary that includes synonyms for each word, as well as the English word’s positive and negative sentiment polarity. Furthermore, SentiWordNet3 accounts for the multi-sense nature of the English language, with each word usage represented by a separate line, and the polar average of multi-sense words was employed in this article.

Senticnet5 has 100,000 English words with sentiment polarity and sentiment intensity. In this article, we primarily used the sentiment dictionary’s polarity category information to create a new sentiment dictionary.

### 4.2. Implementation Details and Metric

We set our model parameters based on BERT-base [3]. We set the dimension of word embeddings and hidden units $d_{model}$ to 768. We set 12 heads for multi-head self-attention. We set the number of layers to 12.

We used the accuracy of sentiment classification as an evaluation metric for text sentiment analysis experiments, i.e., the proportion of correctly classified samples to the overall population. The evaluation metric, accuracy, can be calculated as the following equation:(10)$$Accuracy=\frac{TP+TN}{TP+TN+FP+FN}$$
where TP, FN, FP, and TN refer, respectively, to the number of true positive instances, the number of false negative instances, the number of false positive instances, and the number of true negative instances.

### 4.3. Automatic Evaluation

We conducted comparative experiments between the traditional model and the sentiment temporal analysis model. Table 1 shows that using historical data increases the model accuracy by three percentage points with the pre-trained model, and that using historical data has a significant effect, where BERT-base does not use historical data. By analyzing the historical data, the attention-based sentiment temporal analysis model can extract the user’s sentiment style and the inner pattern of sentiment change, thus improving the model’s sentiment analysis performance. The experimental results show that the addition of domain knowledge can also improve the model’s performance in the sentiment temporal analysis task.

We compared the Attn model with the BERT-base model and the DaAttn model with the DaBERT model. By comparing the results of the two sets of experiments, we can find that the models after using the high-dimensional attention mechanism perform, on average, 2.6% better than the models without the temporal attention mechanism, where the Attn model is 2.7% better than the BERT-base model. The attention-based sentiment temporal analysis model uses the user’s historical posting data. It can extract the user’s sentiment style and the inner pattern of sentiment change by analyzing the historical data so as to improve the model’s sentiment analysis performance. After the above experimental comparison results and rigorous argumentation analysis, we can conclude that this paper achieves the greater advantage and necessity of personalized analysis and attention to the multidimensional temporal sequence.

In Table 2, we report the results on the SST dataset. The table shows that the BERT model has a strong text semantic modeling capability and outperforms deep neural network models based on RNN, BiLSTM, and CNN structures on all four datasets. Furthermore, on different datasets, DaBERT outperforms the original BERT-base, demonstrating that adding sentiment attribute information to words improves the model’s semantic modeling quality.

In the sentence-level sentiment analysis task (i.e., Root mode), the accuracy of DaBERT increased significantly from 90.9% to 92.6% for SST-2 and from 50.4% to 53.9% for SST-5.

In language modeling, DaBERT and BERT-base have the same structure, except that DaBERT adds the network structure for pre-training tasks so that the enhanced model can improve the semantic modeling quality without increasing the model complexity for subsequent target tasks.

Yelp Review-2015 is a larger multi-category dataset that can help to reduce evaluation errors caused by a lack of data volume. As shown in Table 3, when compared to the experimental benchmark model BERT-base, DaBERT performs significantly better in the fine-grained sentiment five classification task, with an improvement of approximately 1.4% in classification accuracy, confirming the importance of sentiment attribute information for sentiment analysis tasks.

We also investigated the validity of the proposed DAM pre-training task and devised the comparison experiments described below.

BERT-base: No pre-training in the target task dataset.BERT-base + only MLM: Only MLM pre-training task is performed.BERT-base + only DAM: Only DAM pre-trainingtask is performed.DaBERT: MLM and DAM pre-training tasks are completed.

As shown in Table 4, both MLM and DAM can improve the original model’s sentiment analysis capability. DAM’s word sentiment attribute information is more important for the sentiment analysis task than MLM’s contextual semantic information. The experimental results also show that the multi-task learning-based pre-training mechanism can combine the benefits of MLM and DAM and add additional domain knowledge information without losing the original semantic information.

## 5. Case Study

We performed a case study for a better understanding of the model performance. In Table 5, we show an example output of our model, where 0 indicates negative and 4 indicates positive. In this case, the BERT-base model, which does not use historical data incorrectly, classifies the second sentence as positive. The original expression means that someone’s hair is on fire and that it looks like a new haircut, while the baseline model ignores historical information leading to misclassification, only focusing on the “new haircut”. In contrast, DaBERT correctly classifies the original content. We infer that historical information plays an important role in sentiment analysis.

## 6. Conclusions

We investigated emotional tracking on time-varying virtual space communication using temporal prediction. Because text data in social networks have temporal properties, we treated users’ continuous postings over time as a time series and combined users’ historical posting records with current postings in social networks for personalized sentiment analysis. This combination of time-varying prediction and sentiment analysis has a broad application value in tasks such as opinion monitoring, financial forecasting, and business decision making.

Based on the BERT model, we proposed a pre-trained language model with domain knowledge enhancement and incorporated the domain knowledge into the original pre-trained model via a pre-training task based on multi-task learning, enhancing the model’s semantic modeling ability and ability to distinguish fine-grained sentiment. We proposed a sentiment temporal analysis model for social networks to improve the prediction of users’ current sentiment status by analyzing temporal information in their historical texts. 

## Figures and Tables

**Figure 1 sensors-22-08450-f001:**
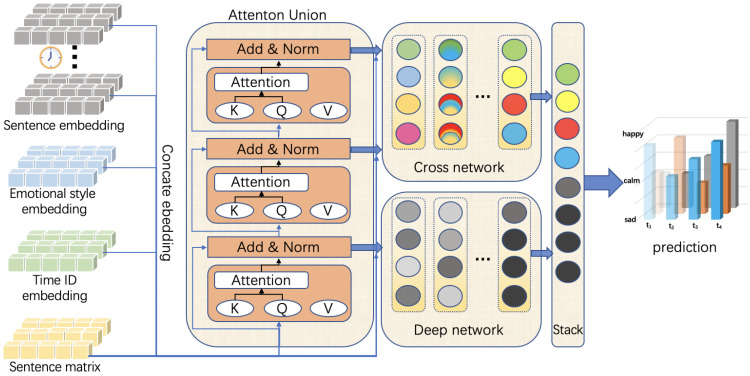
The overview of our proposed model. In the final prediction output stage, the model output is divided into two dimensions. The horizontal axis represents the temporal change from the past to the latest moment in time, and the vertical axis represents the good or bad affective state. In the above figure, we use the three identifiers ‘sad’, ‘calm’, and ‘happy’, which replaced the numerical values, in order to make a simple and easy-to-understand representation. Different colors used in the cross network and deep network represent the fused information obtained after the previous layer.

**Figure 2 sensors-22-08450-f002:**
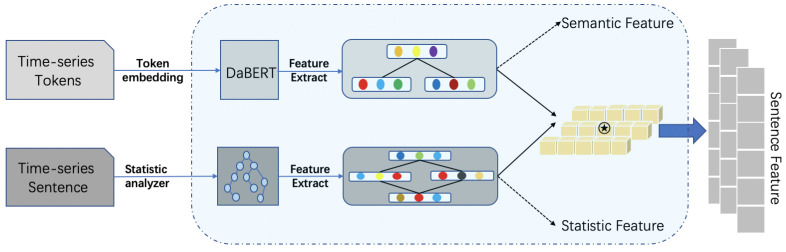
Domain-knowledge-enhanced encoder. The black star in the figure indicates the concatenation of semantic features.

**Table 1 sensors-22-08450-t001:** Twitter sentiment analysis comparison experiment results.

Model	Accuracy
DA-RNN [29]	0.7613
Bi-LSTM [30]	0.7719
EA-LSTM [31]	0.7846
BERT-base	0.8245
DaBERT	0.8296
Attn	0.8471
**DaAttn**	**0.8503**

**Table 2 sensors-22-08450-t002:** Experimental results of sentiment analysis—SST dataset.

Model	SST-2(ALL)	SST-2(Root)	SST-5(ALL)	SST-5(Root)
RNN [32]	83.6	80.2	75.9	41.4
LSTM [33]	85.2	82.6	80.7	42.3
BiLSTM [34]	86.1	84.8	83.5	45.6
CNN [35]	85.7	85.1	84.2	46.5
CNN-LSTM [36]	86.2	85.7	83.9	47.4
BERT-base	92.7	90.9	81.3	50.4
**DaBERT**	**94.8**	**92.6**	**85.3**	**53.9**

**Table 3 sensors-22-08450-t003:** Sentiment analysis experiment results—Yelp dataset.

Model	BERT-base	RoBERTa	LSTM	MultiResCNN	DaBERT
Accuracy	0.9328	0.9331	0.9257	0.9268	**0.9461**

**Table 4 sensors-22-08450-t004:** Experimental results of pre-training task validity analysis.

Model	Accuracy
BERT-base	0.9375
BERT-base + only MLM	0.9389
BERT-base + only DAM	0.9491
DaBERT	0.9567

**Table 5 sensors-22-08450-t005:** Case study.

DaBERT	BERT-Base	Sentiment	Date	Text
0	0	0	2009-05-29 10:56:39	’s hair was on fire right now! Ewww it smells
0	4	0	2009-05-29 11:10:19	Hairspray in hair + lighter&bong = new haircut

## Data Availability

Not applicable.

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
