# Peer review of "Synergy Masks of Domain Attribute Model DaBERT: Emotional Tracking on Time-Varying Virtual Space Communication"

_sensors, 2022, doi:10.3390/s22218450_

Round 1

Reviewer 1 Report

The paper reports on a new method to track users' emotions based on historic data sets instead of a single post.

The paper is easy to read and has an adequate English level.

The authors assume too often that the reader is familiar with some of the concepts that they provide. For example BERT is initially mentioned without any explanation or reference to what it is and why it is relevant for their research. The same for Sentiment140 which is used but not introduced. Regarding the results (i.e. Table 1 and Table 2) provide more details as to which performance metrics were used to calculate the numbers in the table. Link them to the description of the performance metric in the text.

Author Response

Detailed reply to reviewer’s comments:

Comment 1: BERT is initially mentioned without any explanation or reference to what it is and why it is relevant for their research. The same for Sentimen140 which is used but not introduced.

Response to comment 1: We moved the reference to BERT from line 39 to the position where it first appeared in line 38. We added an explanation of BERT in line 164. We have introduced Sentimen140 in line 282.

Comment 2: Regarding the results (i.e. Table 1 and Table 2) provide more details as to which performance metrics were used to calculate the numbers in the table. Link them to the description of the performance metric in the text.

Response to comment 2: We added how to calculate accuracy which is used as an evaluation metric for text sentiment analysis experiments in line 333.

Comment 3: References to the various domains and contents on which this proposal is based should be updated.

Response to comment 3: We agree with the referee that some references should be updated. We have now updated some references which our proposal is based. The corresponding references are:

[5] Poria S, Chaturvedi I, Cambria E, et al. Sentic LDA: Improving on LDA with semantic similarity for aspect-based sentiment analysis[C]//2016 international joint conference on neural networks (IJCNN). IEEE, 2016: 4465-4473.

[6] Lu Y, Mei Q, Zhai C X. Investigating task performance of probabilistic topic models: an empirical study of PLSA and LDA[J]. Information Retrieval, 2011, 14(2): 178-203.

[7] Lin C, He Y, Pedrinaci C, et al. Feature lda: a supervised topic model for automatic detection of web api documentations from the web[C]//International Semantic Web Conference. Springer, Berlin, Heidelberg, 2012: 328-343.

[9] Muhammad A, Wiratunga N, Lothian R. Contextual sentiment analysis for social media genres[J]. Knowledge-based systems, 2016, 108: 92-101.

Comment 4: A specific application case could be included in the ANNEXES.

Response to Comment 4: We added a specific application case in Section 4.4.

We thank the reviewers and remain at your disposal for any further questions.

Reviewer 2 Report

The paper proposes a Sentiment Analysis model based on Time Series Prediction to understand and master the chronological evolution of the user's point of view, considering the extraction of related information from the historical records of publications, and not simply based on the analysis of a single user post, as proposed in existing research.

The introduction is clear in general terms; however, references to the various domains and contents on which this proposal is based should be updated. 

The structure of the work is clear and concise, establishing a series of coherent and measurable objectives.

I believe that the work is highly relevant and brings novelty to the subject of study. Although it solves a technical aspect through a procedure. And in order to open this contribution to a multidisciplinary universe, a specific application case could be included in the ANNEXES.

Minor details:

Figure 1 and Figure 2 

-Review resolution. 

-What is the semantic meaning of the colors used?

For example in Cross Network, Deep Network

Author Response

We added an explanation of semantic meaning of the colors used in line 160. 

We thank the reviewers and remain at your disposal for any further questions.